# Experiences of Using Pathways and Resources for Engagement and Participation (PREP) Intervention for Children with Acquired Brain Injury: A Knowledge Translation Study

**DOI:** 10.3390/ijerph17238736

**Published:** 2020-11-24

**Authors:** Melanie Burrough, Clare Beanlands, Paul Sugarhood

**Affiliations:** 1The Children’s Trust, Neurorehabilitation, Tadworth Court, Surrey KT20 5RU, UK; 2Occupational Therapy Division, Department of Allied Health, Social Care and Advanced Practice, School of Health and Social Care, London South Bank University, London SE1 0AA, UK; clare.beanlands@lsbu.ac.uk (C.B.); p.sugarhood@lsbu.ac.uk (P.S.)

**Keywords:** participation, participation interventions, knowledge translation, environment, acquired brain injury, occupational therapy

## Abstract

*Background:* Children with acquired brain injury experience participation restrictions. Pathways and Resources for Engagement and Participation (PREP) is an innovative, participation focused intervention. Studies have examined PREP in Canadian research contexts, however little is known about implementation in real-life clinical settings. This study aimed to understand experiences of clinicians implementing PREP in a UK clinical context, with a focus on implementation processes and key factors for successful implementation. *Methods:* A qualitative single-site 8-week knowledge translation intervention study, guided by an action research framework, explored clinicians’ experiences of implementation. Six occupational therapists (OTs) working in a neurorehabilitation setting participated. The therapists provided two intervention sessions per week, over four weeks for one child on their caseload. Planning, implementation and evaluation were explored through two focus groups. Thematic analysis was used to analyse data. *Results:* Two themes, “key ingredients before you start” and “PREP guides the journey”, were identified before introducing PREP to practice. Four additional themes were related to PREP implementation: “shifting to a participation perspective”, “participation moves beyond the OT”, “environmental challengers and remedies” and “whole family readiness”. A participation ripple effect was observed by building capacity across the multi-disciplinary team and families. The involvement of peers, social opportunities and acknowledging family readiness were key factors for successful implementation. *Conclusions:* The findings illustrate practical guidance to facilitate the uptake of participation-based evidence in clinical practice. Further research is required to understand aspects of knowledge translation when implementing participation interventions in other UK clinical settings.

## 1. Introduction

Over 1.2 million people suffer brain injuries in the UK annually, with up to 50% of incidences observed in children and young people (CYP) [1]. The most common causes of acquired brain injury (ABI) result from acute trauma, brain tumours, infections, anoxia and childhood stroke [2]. It is estimated that at least 350 children per year in the UK suffer a severe ABI requiring in-patient neurorehabilitation to support recovery [2].

Participation, defined as involvement in a life situation [3] is considered fundamental for children’s development of physical and mental health, happiness and life satisfaction. Eighty per cent of CYP in one study experienced reduced social participation after neurorehabilitation, with all families identifying difficulties with attitudes and social support [4]. Participation following ABI is less frequent when compared with typically developing peers [5], with restrictions in structured community activities, social events, play and household chores [6]. CYP with ABI are more likely to experience participation restrictions due to ongoing physical, communication, emotional and behavioural needs [5], increasing the risk of social isolation and poor health.

Interventions in neurorehabilitation to remedy long-term effects of ABI have traditionally aimed to remediate body functions, attempting to change impairments such as motor, cognitive and sensory deficits [7]. Emerging research however suggests that clinicians working in children’s rehabilitation should primarily offer interventions to improve children’s participation across a range of home, school and community occupations [8]. Attendance in diverse meaningful activities and involvement [9] are key attributes to participation interventions. 

One participation intervention, known as Pathways and Resources for Engagement and Participation [10] (PREP), has shown that children’s participation can be influenced by modification of the environment only [11]. PREP is an innovative participation intervention protocol which encompasses five steps: (1) make goals, (2) make a plan, (3) make it happen, (4) measure process and outcomes and (5) move forward ([10], p.6). 

PREP differs from traditional remedial approaches as it aims to identify strengths and barriers within a child’s natural environment, as opposed to changing underlying impairments such as motor coordination or cognition. PREP offers a practical framework to set participation goals in chosen occupations [10]. A coaching approach is adopted when working with the child and family to agree on an intervention plan with solution-focused strategies to reduce environmental barriers.

Key research offers early evidence for PREP with youth aged 12–18 years old. Two interrupted-time series studies found that following 12 weeks of intervention, goals set in leisure domains using the Canadian Occupational Performance Measure (COPM) [12] demonstrated statistically significant improvements for youth [7,13]. Barriers to goal satisfaction were noted in poor societal attitudes, community opportunities and physical accessibility [13]. A recent formative study also highlighted that not only does PREP support changes in participation, but changes were also observed in motor function, cognition and activity performance [14]. 

Another study examined clinician perspectives when using PREP over a 12-week intervention period, for CYP aged 12 to 17 years old with physical disabilities [8]. Clinicians experienced a new understanding of participation interventions [8]. Notably, therapists *“did not perceive it as "true" therapy if "hands-on" treatment was not provided”* ([8], p. 13,396) questioning the readiness and knowledge translation required when introducing a participation intervention in practice. 

There are no available studies exploring participation interventions in children’s neurorehabilitation, therefore highlighting a gap in practice. PREP offers emerging evidence when working with youth with physical disabilities; however, it has not yet been studied in a real-life clinical context for CYP with ABI. This study therefore aimed to understand experiences of OTs implementing PREP in a UK neurorehabilitation setting for children aged 0–18 years old. Although debate exists around the complex concept of knowledge translation [15], this study assumed a knowledge translation definition of forming partnerships between researcher and participants, with a flow of information exchange [16] to influence evidence-based clinical practice.

Study objectives were to: -Establish planning required before introducing PREP to routine clinical practice;-Implement PREP in a neurorehabilitation setting and evaluate clinician experiences;-Identify key factors that influence PREP implementation.

## 2. Materials and Methods 

### 2.1. Study Design

This study was a qualitative single-site 8-week knowledge translation intervention study, guided by an action research (AR) framework. The study took place in a 25 bed neurorehabilitation setting for CYP aged 0–18 years old with ABI. CYP received a goal-led 24 h rehabilitation programme, supported by an integrated team of professionals, including neurorehabilitation consultants, occupational therapists, physiotherapists, speech and language therapists, psychologists, music therapists, nurses and carers.

A criterion sample of qualified OTs working within the neurorehabilitation setting and treating CYP with ABI were asked to participate. The size of the sample was limited by the total number of OTs working within the setting. All six OTs working within the setting participated. Participants were invited to participate via a letter sent by an independent non-clinical professional. A participation information sheet detailed the research question, aims and methods. Informed consent was obtained by providing OTs with a two-week period before being offered the opportunity to provide written consent to participate. Participants were invited to take part in two focus groups, a follow-up meeting and an intervention phase, over a total period of seven weeks. All six had the right to withdraw from the study at any time. 

The first focus group was conducted initially to explore and prepare for the introduction of PREP to practice. Action planning took place two weeks later, during a follow-up meeting. Initial themes were shared with participants from focus group 1 and group participants designed and agreed upon an implementation action plan (Table A1 and Table A2). The participants selected one child on their caseload to offer two 45 min PREP intervention sessions per week, over a four-week period. PREP intervention was offered as part of routine neurorehabilitation treatment, therefore informed consent from the CYP and families was not required. The CYP selected had already received a multi-disciplinary initial assessment, were undertaking active rehabilitation treatment and were not preparing for immediate discharge home.

Participation goals were set by using either goal attainment scaling (GAS) [17] or the COPM [12] before introducing PREP intervention. Three OTs set a participation goal directly with a CYP on their caseload. Three OTs set participation goals with parents, caregivers and family members as they were unable to directly set goals due to their level of cognitive impairment following an ABI or developmental ability. A second focus group was conducted following the four-week PREP intervention period and evaluated the OT’s experiences. 

### 2.2. Procedures 

AR frameworks assume collaborative approaches and the formation of mutual enquiry [18]. At the time of study, the first author held 12 years of post-qualification experience and was the professional lead, band 8 OT within the service. As the first author also held a caseload and team manager responsibilities, the relationship between the first author and participants needed to be carefully considered. A mutual partnership between the study participants and the first author was sought, aiming to empower participants and reduce perceptions of seniority. With this in mind a professionalising action research framework was selected to guide the process. Professionalising action research ‘seeks improvement in professional practice…on behalf of service users’ ([19], p. 155), whilst promoting partnership between the first author and participants. 

In accordance with professionalising action research, a work-based action research cycle was selected [20], providing a cyclical and reflective framework for PREP implementation (Figure 1). This cycle was chosen as it was developed for use in work-based professional settings. The AR cycle consisted of one preliminary step and four main phases.

#### 2.2.1. Constructing 

One cycle of AR was completed over seven weeks, consisting of four phases. The constructing phase involved participants defining and critiquing participation interventions. PREP was selected as participants felt it provided structured intervention, well-suited to rehabilitation and members had not previously used this intervention. 

#### 2.2.2. Planning Action

The planning action phase lasted two weeks, and involved the first focus group to establish the planning required before introducing PREP to the routine clinical practice. After this two-week period, a subsequent follow-up meeting was held to clarify the focus group findings, share themes and agree on an implementation plan. 

#### 2.2.3. Taking Action 

The taking action phase of the cycle was based on the implementation plan from the first focus group and was completed over four weeks. Two PREP intervention sessions were offered each week for one CYP on each participant’s caseload. PREP was introduced to the multi-disciplinary team via e-mail and during team meetings to familiarise professionals in the wider team with this new intervention approach. During PREP implementation, participants requested peer support, therefore peer group support sessions were organised and facilitated by a clinical researcher, independent of the study. Budget funds were made available for PREP activities. 

#### 2.2.4. Evaluating Action

Finally, the evaluation phase was completed over two weeks. Actions were evaluated through a second focus group involving the first author and all participants. Reflections examined all stages of the action research cycle, implementation of the action plan and participant experiences. 

In children’s occupational therapy there are current challenges—in the generation of evidence-based research and the integration of this into clinical practice [21]. The AR framework therefore provided an opportunity for OTs to translate knowledge into practice and enhance the potential for sustainable change.

### 2.3. Data Collection

Focus groups were deemed suitable for AR to provide joint discussion around shared implementation experiences, whilst triggering critical reflections. The first author assumed the role of focus group moderator and guidelines were agreed to ensure all members adhered to confidentiality. The first author advised that seniority was not considered advantageous and aimed to draw out all of the OTs during discussions. The first focus group, lasting 1.5 h, included five semi-structured questions to explore the introduction of PREP to practice (Table A3). The focus group questions were adapted from an existing research study exploring clinician perspectives of using PREP within a research setting [8], in order to reduce moderator leading questions and allow for probing. 

The second focus group used nine semi-structured questions (Table A4) to evaluate participant experiences of implementing PREP in practice. The second focus group lasted 2 h, and questions were again adapted from existing research [8] with a focus on implementing PREP in a neurorehabilitation setting, evaluating the OT’s experiences and identifying factors that influenced PREP implementation. 

The findings were digitally recorded and then transcribed verbatim by the first author. Confidential data such as child or organisational names were replaced with pseudonyms. All digital recordings were transferred and stored on an encrypted PC to comply with data protection principles.

### 2.4. Data Analysis 

The content of each transcript were read and re-read by the first author to increase familiarisation, note initial ideas and search for patterns. Braun and Clarke’s six step thematic analysis [22] was applied to analyse focus group transcripts. To gain an in-depth understanding of the data, transcripts and initial codes were derived by the first author, highlighting data relevant to the research question across the whole data set and collating particular quotes that were relevant to each code. Complete coding was undertaken to ensure relevant words and phrases were coded. At this point, transcripts and initial codes were shared with the second and third authors to check independently of the first author. All authors then searched the codes for potential themes, drawing data together which was relevant to each suggested theme. 

The next stage involved drawing together a thematic map for themes. The initial themes derived from focus group 1 were shared with the participants during the follow-up meeting, providing an opportunity to member check themes and further refine themes. Further defining and naming of themes from both transcripts took place with the second and third authors, with discussions around definitions for each overarching theme. During the six-step analysis, themes and subthemes were reflected upon, checked with the original transcripts and analysed with direct quotes to ensure that thematic mapping derived meaning from the entire data set.

Data were collected from the criterion sample at set points during the AR cycle. This did not allow for continued recruitment or data collection until the concept of saturation could be achieved. However, the in-depth focus group discussions using open questions created sufficient data to gain a plausible understanding of the issues. The iterative nature of data collection and analysis allowed for detailed exploration of themes as they emerged and developed over the course of the study.

### 2.5. Study Rigour 

As the aims of this study were to implement PREP in this particular neurorehabilitation setting, the findings cannot readily be generalised to different population groups. No exclusion criteria were set and all OTs regardless of gender, ethnicity, age and experience were eligible. The study sample was limited by the number of occupational therapists working within the setting.

The study established trustworthiness through principles of credibility and transparency by member checking, following the first focus group. The project was time limited, therefore themes from focus group 2 could not be shared with participants in the same way as focus group 1. Triangulation was considered in gaining a variety of participant views, although increasing findings through representation of different data collection methods and study co-design was not possible due to time constraints.

The first author was aware of the close connection to participants during routine clinical practice, throughout each stage of the AR process and during data analysis. In routine clinical practice the first author also provided support and supervision to the team of OTs, which may have influenced study findings. As part of the implementation action plan participants identified the importance of peer support during the taking action phase. A clinical researcher, independent from the study, facilitated peer support groups during the action phase, which provided a space for reflective thinking, without the influence of the first author. The first author and clinical researcher met to reflect on the peer group sessions before the second focus group, giving the first author an external perspective of the taking action phase.

A reflective diary was completed by the first author throughout the study to increase reflexivity [23]. Reflective diary themes considered the potential influence of the first author and the participant relationship on study findings. Themes highlighted the need to draw out all participant views during focus group discussions. The first author reflected on the routine responsibility of professional lead OT, whilst balancing the role of focus group moderator. Diary experiences and reflections were shared and discussed with the other authors to increase transparency. This supported the first author with allowing for enough time and space to draw out all participant views and experiences.

### 2.6. Ethical Considerations

Ethical approval was granted from the School of Health and Social Care Ethics Committee at London South Bank University on 10th May 2017, study number 17/A/32. Permission from the organisation’s research board was given. All participants gave their informed consent for inclusion before they participated in the study. This study was classified as a service evaluation; therefore, Health Research Authority approval was not required.

## 3. Results

### 3.1. Sample Characteristics

Six OTs participated in the study, with varied levels of experience and seniority. Table 1 outlines participant characteristics.

### 3.2. Emerging Themes 

Six themes were identified from the data. Two overarching themes related to establishing the planning required before introducing PREP to routine clinical practice and four themes related to implementing PREP and evaluating OT experiences of implementation. Each theme will be reported on in turn. All names have been replaced with pseudonyms.

Two themes, before introducing PREP to practice, included: “key ingredients before you start” and “PREP guides the journey” (Figure 2). Four additional themes were related to PREP implementation: “shifting to a participation perspective”, “participation moves beyond the OT”, “environmental challengers and remedies” and “whole family readiness” (Figure 3).

### 3.3. Key Ingredients before You Start 

An overarching theme emerged from the planning stage of the action research cycle when preparing to implement PREP for the first time: *key ingredients before you start.* Key ingredients in preparation for PREP use were suggested to set and work on one participation goal at a time, engage outcome measurement and build a participation team.

One key ingredient was recognised as working on one participation goal at a time. This took a different direction than usual routine practice, which involved working on multiple activity focused goals. Sarah highlighted:
“You get to the end of a placement where you feel like you have moved ever so slowly or not at all in these large number of (goal) areas.”

Sarah went on to say that:
“One participation goal might be greater than helping the family move forward with five or six.”

Using PREP intervention to work on one participation goal at a time appeared to offer opportunity for focused, high intensity therapy to increase children’s participation. Katy suggested:
*“It drives that intensity doesn’t it? in terms of intervention, which we know people need, but if you’re covering a number of goals, how do you get that intensity of intervention, you know giving lots of repetition, lots of practice.*”

Katy’s view of using PREP to work on one goal at a time supported the need for children to receive intensive, repeated input when receiving participation focused intervention.

Similarly Hannah gave examples of working on participation goals such as visiting the local park with family. Hannah commented that PREP intervention is:
*“Very clear focused, repetitive and you make progress.*”

Working on one participation goal was recognised as needing early prioritisation, to guide therapist and family focus during the rehabilitation journey. Another key ingredient to prepare for PREP introduction was considered to be engaging in the use of outcome measurements such as the COPM. The engagement of outcome measurements supported OTs with underpinning changes in participation goals. Sarah reflected when preparing outcome measurement for PREP intervention it enables OTs to:
*“Stay focused, do the COPM, that’s your core thing.*”

Sarah also described that the suggestion of completing COPM more frequently
*“Felt like a big shift.*”

However, there were different feelings about how frequently outcome measurement was required when preparing to introduce PREP to the CYP and families. When considering outcome measurement using the COPM, Beth described frequency of measurement as:
*“Not as regularly as twice a week, parents would find that too much*”

In contrast Emma felt that when introducing PREP intervention, outcome measurement should:
*“Be more than once a week.*”

There appeared to be consensus when engaging in outcome measurement before PREP introduction; however, OTs felt that they required different time frequencies in outcome measurement according to the individual CYP and family needs.

Prior to PREP implementation, the OTs recognised the need for the key ingredient of building a participation team. It was suggested that an integrated approach, drawing on members of the multi-disciplinary team, the child, family and supports in the community (for example sports coaches) was needed. Alice commented that there should be:
*“Shared responsibility for a participation team” *and this needed to be developed during implementation* “because I feel like the understanding and that shared responsibility of the participation team isn’t there yet.*”

Communication and collaboration appeared to form initial building blocks for a participation team.

### 3.4. PREP Guides the Journey

The PREP manual was seen as a map and practical guide to keep the OTs on track. Sarah outlined the PREP process as:
*“You set goals, you make a plan, you make it happen and then you check it.*”

Before implementation, the PREP process was considered as a tool for keeping OTs focused by working through goal setting and treatment planning logically. Katy highlighted:
*“If you do work through the resources you’ve got in this you’ll be looking at it thinking oh I actually haven’t done what I’m meant to be doing this week, I’ve got to keep on this, this is my plan, I’m using this tool, you’ve got something really specifically to focus you.*”

Beth commented that PREP will *“structure the plan of intervention”* and reported *“if someone else needs to follow the process that you have done, it starts to kind of break down the different components.”*

It appears that PREP offers a concise map to form a measurable treatment plan, with the opportunity to involve different members of the multi-disciplinary team to follow the process if needed.

Finally, in order for PREP to guide the journey, it was felt that PREP should be introduced at the start of a rehabilitation journey to clarify family expectations. Beth highlighted that early expectations of neurorehabilitation programmes can include a focus on *“walking and talking.”* At the start of rehabilitation Hannah commented:
*“They’re not used to thinking about participation yet, helping them to understand that. If you’re going to the park, they’re not just going to the park for fun, it’s showing them why we’re doing that and the skills we’re using and how that’s rehab.*”

Introducing PREP early on appeared to introduce participation goals and intervention early on in the rehabilitation journey. Alice however recognised that participation may have changed and questioned whether the child would be: “*prepared to take on or participate in something in not its true or original form.”*

Four themes were identified when implementing PREP and evaluating OT experiences of implementation: *shifting to a participation perspective**; participation moved beyond the OT; environmental challengers and remedies; and whole family readiness* (Figure 3).

### 3.5. Shifting to a Participation Perspective

Adopting PREP intervention meant letting go of traditional, remedial therapy approaches focused on component skills or impairments in mobility or cognition and shifting to a participation perspective. For some therapists PREP offered an intervention approach to focus on participation. Sarah illustrated:
“PREP kind of underpins why we’re in this job because you want people despite health problems to enjoy life and participate in what life has to offer…”

Emma recognised that PREP offered a new, flexible approach to work on participation even when a young person may not have achieved skill mastery:
“Playing play station 4…he’s indicated that he would really like just to try and see what he can do, I probably wouldn’t have thought about doing that because I know it’s going to be really really difficult.”

For others however shifting to a participation perspective felt anxiety provoking, particularly when attempting coaching to empower families to solve problems. Beth reflected on her own *“hesitations”* and Alice highlighted:
*“Not having experience of coaching techniques, of managing difficult situations or having those tricky conversations with parents, because I don’t have that knowledge and experience I think is probably why I didn’t do some much of that…*”

Therapists identified that peer support, facilitated by an experienced therapist would address feelings of apprehension and changes to intervention practices.

### 3.6. Participation Moved Beyond the OT

PREP built capacity with families and the multi-disciplinary team by using the pathway to form solutions independently of the therapist. Not only did the therapists feel that young people demonstrated more insight into participation challenges than they anticipated, some therapists felt that CYP and families began to generalise problem-solving techniques to other participation opportunities. One young person initially achieved his participation goal of going to a local fast food outlet with his family. Soon after this the young person and his family identified other community experiences that they wanted to achieve whilst undergoing neurorehabilitation. Alice reflected:
“I was very taken aback by how he had come up with all of these strategies, it was so important that they’d come from him. He’s gone to the bakery, to the harvester (restaurant), to the seaside at weekends. His Mum and sister have done it, they’ll say ‘well where shall we go next and then he comes up with the next idea and then what do we need to do? oh well I need to be able to walk outside, I need to be able to stand up to put the pennies in the slot machine you know he’s coming up with those things.”

Beth discussed a participation goal around a Father and young person making a train journey from the rehabilitation centre to home:
*“I went and spoke to Dad about if I was doing that trip what I would do and I would look for. Dad took that on board and I spoke to him yesterday and he said ‘yea it was fine’ he already knew that (name) would know the way home and was physically able to do it but his concerns were still (name’s) behaviour and communication difficulties.*”

Beth went on to describe how she developed a strategy with this young person:
*“Maybe you could learn how you say I have difficulty with talking and he said words are hard so he’s made his own little script for that.*”

Once therapists shared knowledge of local community activities and leisure opportunities, CYP and families appeared to increase their active involvement in participation experiences. After one CYP achieved his original participation goal, he started to apply his skills to familiarise himself with car journeys to participate in community outings, Alice described:
*“They built up their car parking, car driving practice, they built up every night by themselves, they visited local areas*”

Therapists perceived attitudinal changes of professionals during increased young person and family involvement, leading to greater ownership and shared management of participation challenges. 

Through the development of a shared participation vision and action plan, PREP was perceived by one therapist as causing “*ripple effects” (Sarah).* To illustrate, Emma highlighted:
*“I had some positive engagement with care staff who identified there were some things they could do with the young people outside of sessions that they wouldn’t have done otherwise.*”

In some cases, members of the multi-disciplinary team began contributing to PREP planning and used strategies without OT involvement. Ripple effects were seen not only for CYP and families but also reaching wider members of the multi-disciplinary team. 

### 3.7. Environmental Challengers and Remedies

A number of environmental factors were recognised as challengers and barriers to successful participation. Hannah recognised that PREP *“helped get that real participation goal, think about the barriers and facilitators and steps towards it.”* Most barriers were perceived to be physical environmental restrictions including ‘noisy environments,’ ‘wheelchair accessibility’ and ‘the temporary nature of the rehabilitation centre’. Other barriers however reflected social factors such as ‘his friends aren’t here,’ whilst other CYP were worried about societal reactions to communication difficulties following a brain injury.

Some environmental factors were observed to remedy participation challenges. Social and familial relationships appeared to hold great importance in remedying participation challenges. For one young person, social engagement appeared to hold greater importance than the activity itself, as Alice suggested:
*“Riding your bike is not the meaningful part, it was more about doing it, that feeling of belonging, being with the people that they wanted it to be with, that sense of socialisation in the way that he wanted it to be.*”

Being with family and peers was perceived as crucial for this young person to feel valued and involved:
“It was that he wanted, to go with his Mum and his sister, he wanted to be able to have that whole experience, it wasn’t about could he order, could he eat it safely, could he sit in the car, he wanted to tell Mum off every time that she sung in the car on the way there.” (Alice)
Although some CYP were unable to engage with peers in their typical home, school and community environments, they often wanted to participate with new peers during rehabilitation. Sarah illustrated that one young person reported “*no, no we really want to do it together.”*

### 3.8. Whole Family Readiness 

The concept of family readiness influenced therapist abilities to implement PREP. Notably, family anxiety and stress were acknowledged as key influencers to engagement in participation, goal setting and finding solutions. For instance, Emma highlighted:

*“The families have got so much going on, the family I worked with they’ve got lots of other anxieties and worries, for them to try and focus and think about something else was quite hard.*”

Factors such as re-housing, changes in schooling and the young person’s mood were identified as areas of consideration before introducing PREP. Changes to participation patterns and loss were also acknowledged. Some young people wanted to participate *“when they were better,”* whilst others were aware of *“who they were and now that’s different” (Emma).* Alice reflected on perceiving children’s participation in a new way after brain injury:
*“Whether the child is going to be able to participate in the way they did before and if they can’t are they happy to accept the change in how they participate? That wasn’t what he’d done before so in his eyes well that wasn’t achieving it.*”

Contrastingly, resilience and strong family networks appeared to reduce family anxiety and stress to increase readiness for PREP intervention. One family had another child with a disability and Sarah found that they drew on past experiences to overcome participation challenges:
*“The family had their own experiences prior to injury with another sibling in the family who has special needs, and so adjustment to their young boys’ brain injury was very different to other families. Their acceptance of needing to do things differently, to be innovative about the way you do things, their personal circumstances have really meant that they don’t have that as a barrier, adjusting to disability.*”

## 4. Discussion

This study aimed to understand how OTs implemented PREP, a participation intervention, in a UK clinical context. The study explored planning required before introducing PREP to routine clinical practice and evaluated the OT’s experiences of implementation when working with CYP aged 0 to 18 years old with ABI in a neurorehabilitation setting. Study findings highlight important messages for practice, when introducing a participation intervention for the first time.

Several key ingredients were acknowledged in order to introduce PREP to practice, notably setting and working on one participation goal at a time, engaging outcome measurement and mobilising a participation team. Enabling the child and family to form a team to work on one participation goal appeared to increase the intensity and focus of the intervention. Notably active ingredients such as caregiver support and a supportive environment for the child have previously been recognized to improve CYP participation outcomes [24], echoing the value in identifying a supportive participation team to work on the goal with the child.

Implementation of PREP initiated new directions in practice. Although participation outcomes were enhanced, participants often felt anxious when adopting new ways of working, sometimes feeling that they may overlook rehabilitation goals in activity or body functions and structure domains [3]. Reluctance to shift to a participation perspective is consistent with previous research, challenging the nature of the environment and therapeutic need to work on personal factors [8]. Two practical steps were recognised to support early adoption of PREP. Firstly, peer support offered space for reflection, problem solving and sharing PREP strategies. Additionally, training on coaching techniques was suggested to enhance knowledge in order to explore participation challenges.

PREP enabled extension of knowledge and built capacity with others. Knowledge sharing allowed CYP and families to generalise problem-solving techniques to new participation challenges. One young person successfully achieved his participation goal of going to a local fast food outlet, through developing participation strategies with his therapist. Following this success he then participated in other community experiences such as visiting the local bakery, restaurant and seaside without input from his therapist. Capacity building was also seen within the multi-disciplinary team. Another participant commented that care staff continued with participation strategies outside of intervention sessions. PREP was described as causing “ripple effects” suggesting that immediate successes grew outwardly with the support of the identified participation team. The participation team and knowledge sharing appear to be fundamental for successful implementation.

PREP was observed to prepare CYP for transition between neurorehabilitation and discharge to their local home, school and community. Previous research has shown that parents require support to build confidence in managing a range of complex difficulties following neurorehabilitation [25]. This study’s findings suggest that PREP offers promising findings to equip families to support self-management of participation challenges following discharge. This approach is congruent with person-centered care principles of empowering families to self-manage health needs following illness [26]. Focusing on participation goals for discharge could therefore offer long-term quality of life and health benefits to families. Participation interventions appear to address the need for personalised care, which may result in a lesser need for intensive professional input in the community.

The environment was recognised as a salient factor during PREP intervention. Although environmental restrictions were experienced, environmental remedies were recognised as facilitators. Physical and social barriers included feelings of anxiety when leaving the neurorehabilitation centre, combined with unfamiliar and noisy environments. Existing peer relationships were not always present post ABI due to geographical distances, limiting support when working on participation goals. Once environmental challengers were identified, strengths could be drawn on to overcome these barriers. In this study feelings of belonging and engagement often overcame participation barriers. Some CYP requested to work on participation goals with peers also receiving neurorehabilitation. One CYP reported that, when bike riding, being with friends and socialising was more important than the activity itself. Another CYP visited a local restaurant and reflected that spending time with family and singing during a car journey made all the difference to his happiness and enjoyment. Identifying and drawing on strengths within the social and familial environment appeared to remedy initial participation challenges.

It is noteworthy that even though social skills may be disrupted or impaired post ABI, social experiences for young people receiving neurorehabilitation were most significant in remedying participation difficulties. Understanding social norms and boundaries can be challenging following ABI [27] however social opportunities and peer support were key participation influencers. Findings show that peer support can remedy participation challenges due to benefits of enjoyment, socialisation and the sense of belonging. In neurorehabilitation specifically, therapists may consider implementing participation interventions involving existing peer groups or creating opportunities for new friendships with peers who have experienced ABI.

Finally, family readiness was integral before PREP introduction. Family factors including anxiety, stress, mood and life changes such as re-housing and schooling were identified. Some families identified life adjustments, reflecting that aspects of participation may be lost or changed. Literature highlights that many caregivers experience grief, loss and family strain following ABI, often requiring a period of adjustment to their child’s disability [28]. Conversely, one family drew on past experiences of supporting a sibling with a disability to help them overcome new challenges, increasing readiness for PREP intervention and reducing family anxiety. Consequently, when introducing PREP therapists recognised the need to actively listen to the family and CYP during their rehabilitation journey, whilst acknowledging difficulties along the way.

### 4.1 Key Implications for Practice

Key guidance and recommendations were identified for sharing application of knowledge when introducing PREP to clinical practice:Key ingredients including ‘working on one participation goal at a time, engaging outcome measurement and building a participation team’ can prepare therapists to introduce a participation intervention in a neurorehabilitation practice setting.Peer support and formal training around parent coaching should be offered to support clinicians with adopting PREP intervention.Sharing knowledge of problem-solving and participation strategies will enhance capacity building of others, such as the multi-disciplinary team, parents, caregivers and the child or young person to increase opportunities for participation success.Participation interventions are perceived as valuable by OTs to prepare CYP for participation in their local home, school and community following discharge from neurorehabilitation.Opportunities for socialisation with existing peer groups or peers who have experienced ABI should be created to support benefits of enjoyment and belonging when implementing participation interventions.Family readiness is a key factor to consider when implementing participation interventions.Therapists should actively listen to and acknowledge family difficulties during the rehabilitation journey to help the family plan for achievable and client-centred participation interventions.

### 4.2 Limitations and Future Direction

This study contributes to a knowledge gap by offering guiding principles of how OTs can facilitate the uptake of PREP in a children’s neurorehabilitation setting in the UK. Many earlier studies considered PREP use with youth. Key implications for practice from this study were considered for CYP ranging from 0–18 years old, however further studies may offer more specific guidance to differentiate recommendations for PREP use between younger children and youth. 

Although key ingredients have been proposed for the introduction of PREP to clinical practice, they may not be transferable for use in other areas of paediatric occupational therapy practice. It would be valuable for future studies to examine perceptions of the proposed key ingredients to investigate transferability. Further action research cycles could offer the opportunity to evaluate key ingredients to introduce PREP to practice in different paediatric practice settings in the UK. 

As this study did not examine the perspectives of CYP and families, future qualitative research would add further triangulation of views and experiences. Furthermore, a longitudinal study would be of interest to follow-up CYP participation experiences following discharge. It would be useful to understand whether families benefited from PREP intervention following discharge, and whether or not participation strategies were effective in home, school or community environments. 

Finally clinicians would benefit from further practice guidelines to support implementation of participation interventions. This study somewhat offers a starting point for clinicians working with CYP who have experienced ABI to offer participation interventions in their clinical practice.

## 5. Conclusions

This study examined how occupational therapists can introduce and implement participation interventions in a children’s neurorehabilitation setting. The study specifically examined PREP, a participation intervention aiming to improve children’s participation through the identification of environmental barriers and facilitators [10]. This knowledge translation study considered knowledge and application to clinical practice to further contribute to the evidence base for participation focused interventions.

Findings offer practical principles to apply knowledge when supporting early adoption of participation interventions in practice, whilst building capacity to support generalisation of PREP strategies beyond therapist led intervention. The involvement of peers, social opportunities and acknowledging family readiness were key factors for successful implementation. Therapy-led peer support and training in coaching were identified to remedy challenges when adopting a new participation perspective and directions in practice. 

## Figures and Tables

**Figure 1 ijerph-17-08736-f001:**
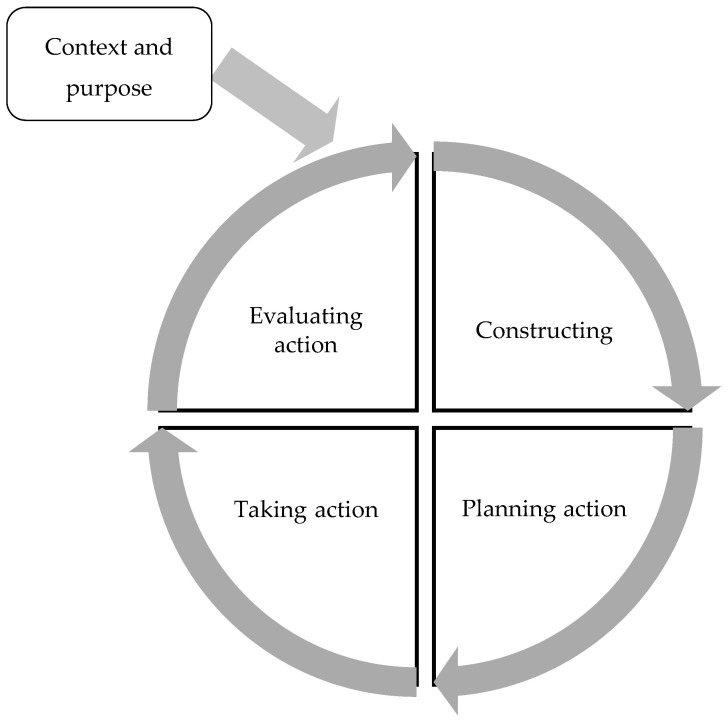
Action Research Cycle adapted from ([20], p. 9).

**Figure 2 ijerph-17-08736-f002:**
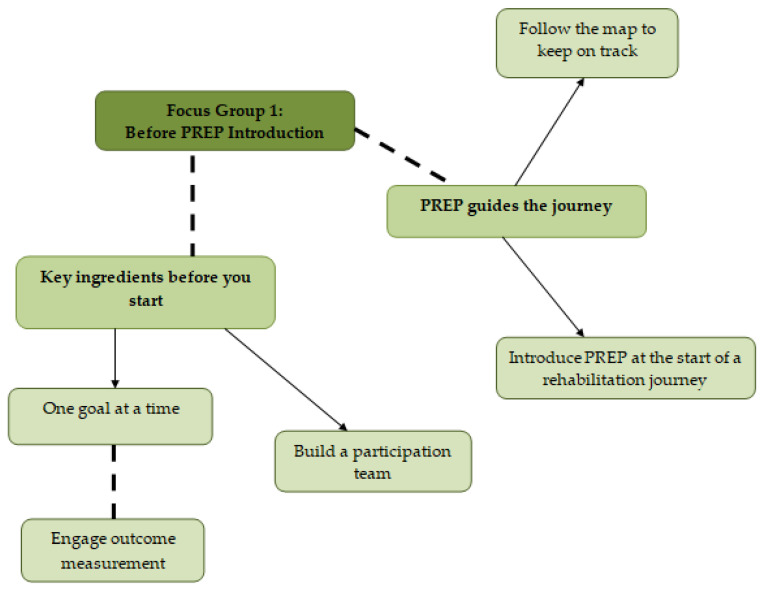
Structure of themes generated by the data in focus group 1: before PREP introduction.

**Figure 3 ijerph-17-08736-f003:**
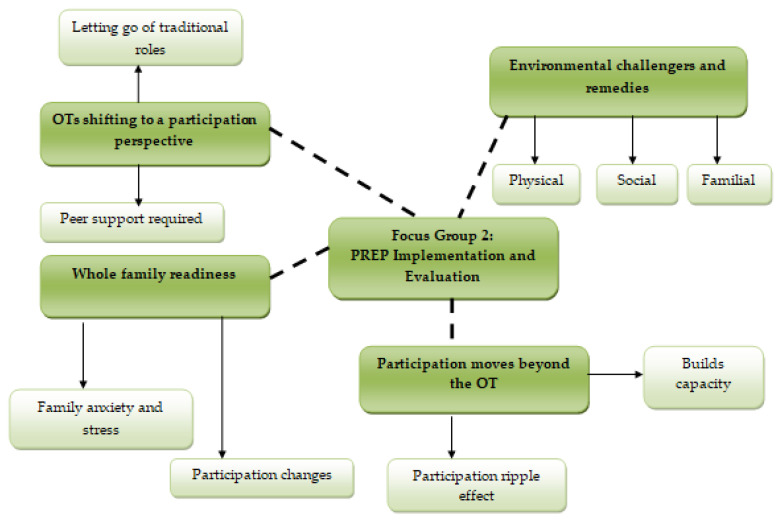
Structure of themes generated by the data in focus group 2: PREP implementation.

**Table 1 ijerph-17-08736-t001:** Sample Participant Characteristics.

Participant Characteristics	Number
**Gender**	
Male	0
Female	6
**Experience in Clinical Practice**	
0 years up to 2 years	0
3 years up to 5 years	1
6 years up to 8 years	1
9 years up to 11 years	1
12 years up to 20 years	2
21 years or more	1
**Level of Seniority (according to agenda for change banding scale)**	
5	0
6	2
7	3
8	1
**Occupation**	
Full time occupational therapist	3
Part time occupational therapist	3

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
