# Peer review of "Experiences of Using Pathways and Resources for Engagement and Participation (PREP) Intervention for Children with Acquired Brain Injury: A Knowledge Translation Study"

_ijerph, 2020, doi:10.3390/ijerph17238736_

Round 1

Reviewer 1 Report

Thank you for the opportunity to review your interesting work. I have a few questions and comments for consideration.

Numbers at the beginning of sentences need to be in words, not figures.

Were all goals set with child/young person or were some set with parents?

The relationship between researcher and participants should be described and potential influences reflected upon and presented.

You need to support the use of thematic analysis with an appropriate reference e.g. Braun and Clarke, they are in the reference list but not cited in text.

There were 6 participants but only direct quotes from 3 people are reported(Sarah, Alice, Emma). Are the views of all participants represented in the themes?

I wish you luck with your future work.

Reviewer 2 Report

This is a well-conducted study identifying the involvement of peers, social opportunities, and family support as crucial factors for successful Pathways and Resources for Participation and Engagement (PREP) implementation among brain-injured children. I do not have any major concerns. I would recommend the authors add a section, where applicable, the beneficial implementation for participation interventions in youth or young adults to help guide the reader on age-specific follow-up.

Reviewer 3 Report

Thank you for this interesting study.

However, I have some comments on this manuscript.

There are many abbreviations in the manuscript. It would be valuable with an abbreviation list.

Six participants can be enough, if they give rich data. However, I lack information about how the researchers evaluated the saturation of the data. I lack information about selection of sampling method. Was it a convenient sample?

When were the data collected? The ethical approval was given 2017.

I would like more details in the data analysis. Who did the analysis, and how was the transcripts done?

Reviewer 4 Report

- The purpose and results of the study are not matched. The purpose and results of the study are not matched. The objectives and results of the research cannot be matched. Author need to modify one of them to match.

- Line 103: What do you mean (p.155) was selected as a ~~?

- In interview, the information about participants is very important. More specific information about OTs should be provided. Also, more detail information about interviewer should be descripted.

- PREP intervention is for children with acquired brain injury. I thought that authors provided information about children with acquired brain injury in this study because characteristics of children might effect on intervention effect or implementation.

- Study Design might be revised. Action Research Cycle could be described in procedures. The study design in this manuscript was qualitative study because authors used focus groups interview.

- Specific semi-constructed question should be provided.

- Regarding to data analysis, number of data coders, description of the coding tree, derivation of themes, software should be provided.

- The code of sentences which stated in the results section should be provided.

Round 2

Reviewer 3 Report

The authors have revised the manuscript according to previous comments. I think it is suitable for publication.

Reviewer 4 Report

Dear

Thank you for your opportunity to review this manuscript. 

This manuscript was revised according to my comments. 

I appreciate for your effort.